# Associations of Morphological Changes in Skeletal Muscles of Preschool Children in China Following Physical Activity

**DOI:** 10.3390/children10091538

**Published:** 2023-09-11

**Authors:** Pengyu Deng, Hayao Ozaki, Toshiharu Natsume, Dandan Ke, Dajiang Lu, Koya Suzuki, Hisashi Naito

**Affiliations:** 1Graduate School of Health and Sports Science, Juntendo University, Chiba 270-1695, Japan; ko-suzuki@juntendo.ac.jp (K.S.); hnaitou@juntendo.ac.jp (H.N.); 2Institute of Health and Sports Science & Medicine, Juntendo University, Chiba 270-1695, Japan; ozaki.hayao@gmail.com (H.O.); kedandan@fudan.edu.cn (D.K.); 3School of Sport and Health Science, Tokai Gakuen University, Miyoshi 470-0207, Japan; 4Department of Human Structure & Function, Tokai University School of Medicine, Isehara 259-1193, Japan; natsumetoshiharu@gmail.com; 5School of Public Health, Fudan University, Shanghai 200433, China; 6School of Kinesiology, Shanghai University of Sport, Shanghai 200438, China; ludajiang2000@aliyun.com

**Keywords:** physical activity, growth and development, muscle growth, skeletal muscle

## Abstract

Purpose: Physical activity (PA) is likely to be the most important modifiable factor in skeletal muscle development. However, the influence of PA on the skeletal muscle of preschool children has not been thoroughly investigated. The main objective of this study was to quantitatively measure PA, and then, to assess whether associations exist between site-specific muscle changes and PA in relation to sex and weight statuses in preschool children aged 3 to 4 years. Methods: A total of 86 healthy preschool children, aged 3–4 years, were instructed to wear an accelerometer for seven consecutive days. The number of steps taken daily, and minutes spent in moderate–vigorous PA (MVPA) and total PA (TPA) were recorded. Muscle thickness was measured by B-mode ultrasonography using a 5–18 MHz scanning head. Muscle thickness was measured at seven sites: the lateral forearm, upper arm, abdomen, anterior and posterior thigh, and anterior and posterior lower leg. Results: There was no significant difference between boys and girls in terms of MVPA and TPA on weekdays and weekends. According to the linear regression models, after adjusting for daylight duration, the muscle of the posterior thigh was significantly positively associated (*p* < 0.05) with daily steps and MVPA on weekdays for boys and girls, respectively. Conclusions: We found that the muscle thickness of the posterior thigh in preschool children was significantly positively associated with PA, as measured by daily steps and MVPA. We suggest that for the overall health and well-being of preschool children, the levels of PA should be maintained and/or increased, and preferably transformed into a regular part of daily living.

## 1. Background

Childhood is a key period for skeletal growth, whereby the body increases in size, yet, notably, the changes in the proportions of muscle and fat mass are influenced by gender. Previous studies have reported differences in skeletal muscle and fat mass in children [1] and gender-based disparities have also been documented [2,3]. However, evaluation of the results of muscle thickness in the forearm, thigh, and lower leg showed no significant differences by sex in children and adolescents aged 0–18 years [4]. It is difficult to measure skeletal muscle thickness in children and adolescents, partly because the site of the examinable skeletal muscle can only be located in the upper or lower limbs.

Skeletal muscle is highly plastic and is capable of responding to a variety of stimuli [5,6]. Over the past few decades, the quadriceps muscle group has frequently served as an indicator of age-related alterations in muscle and lower limb strength [7]. Several studies have reported that daily PA provides an important stimulus to the musculoskeletal system, which can improve muscle development. Some studies have suggested that children and adolescents with higher fat masses have lower-than-expected daily PA and are at an increased risk of fractures [8,9]. Growing evidence suggests that daily PA in children and adolescents is negatively associated with total body fat [10,11]. Therefore, additional studies are needed to better understand the association between daily PA levels and body fat percentages, which will provide useful insights for interventional studies that focus on enhancing daily PA and reducing the body fat percentages in the target group.

Recent research showed that reduced PA levels are a strong predictor of weight gain and/or obesity in children and are associated with the risk of cardiovascular disease factors [12,13]. In addition, the role of muscle thickness has been increasingly recognized in the prevention of chronic disease in adults [14], and the features of metabolic syndrome have been negatively associated with body fat percentages in men [15] and women [16]. Obesity is associated with an increase in muscle mass [17], suggesting that the strength production capacity of obese individuals is higher than that of non-obese participants. However, because obesity is also associated with a decrease in muscle quality [18,19] and a higher body fat percentage owing to less physical activity [20], the muscle gain yielded may eventually be of too poor quality to offset the increase in body fat percentage associated with reduced PA [21]. It is important to understand whether the relationship between PA and muscle thickness identified in children is independently related to weight status in children, in order for effective evidence-based prevention and treatment strategies to be implemented as early as possible.

We previously reported that the quantity and intensity of PA is predominantly related to weight status in children [22], yet it is unclear whether preschool children are also associated with PA and skeletal muscle. Therefore, further studies would be beneficial to determine whether daily PA and/or muscle thickness could be proposed as health markers at these ages. In view of the above, the main objectives of this study were to:(1)Evaluate the skeletal muscle thickness and objectively measure PA;(2)Assess whether associations exist between site-specific muscle thickness and level of PA with regard to sex and weight status in preschool children aged 3 to 4 years.

## 2. Methods

### 2.1. Participants

A total of 113 healthy preschool children, aged 3–5 years were selected, using the selection criteria for admission into the study: attendance at a preschool, permission from parents to participate in this study, and no injuries or current illnesses reported. The parents of each subject provided informed consent in line with research protocols adhering to the Declaration of Helsinki, which was approved by the ethics committee of Juntendo University.

### 2.2. Anthropometrics Data

In 2017, all anthropometric data were collected by trained staff under the supervision of a school nurse. Height and body weight were measured using a portable stadiometer and portable digital scale (TCS-200-RT; Yao Yi, Shanghai, China), respectively. Weight status (i.e., underweight, normal weight, and overweight/obese) was determined according to the age- and sex-specific cutoffs of the Working Group on Obesity in China (WGOC) [23]. Table A1 describes BMI standard cutoff criteria for the overweight and obese classification of preschool children in China.

### 2.3. Muscle Thickness

Muscle thickness and fat thickness (FTs) were measured via B-mode ultrasonography using a 5–18 MHz scanning head (Noblus; Hitachi, Tokyo, Japan). The scanning head was prepared using a water-soluble transmission gel that provided acoustic contact without depressing the skin surface. MTs and FTs were obtained at seven sites on the anterior and posterior surfaces of the body [24]. The seven anatomical landmarks for the chosen sites were defined as follows: the upper arm (UA) site was approximately 60% distal to the entire upper arm between the lateral epicondyle of the humerus near the elbow and the acromial process of the scapula at the shoulder; the forearm (FA) site was on the anterior surface, 30% proximal to Y between the styloid process of the wrist and the head of the radius at the elbow; the abdominal (AB) site was at a distance of 3 cm to the right of the umbilicus; the anterior thigh (AT) and posterior thigh (PT) sites were on the anterior and posterior surfaces of the upper leg, midway between the lateral condyle of the femur at the knee and the greater trochanter at the hip; lastly, anterior (AL) and posterior (PL) lower leg sites were approximately 30% proximal to the talus between the lateral malleolus of the fibula at the ankle and the lateral condyle of the tibia at the knee.

### 2.4. Physical Activity

We measured PA using a uniaxial Kenz GS AC (Lifecorder, Suzuken Co., Ltd., Nagoya, Japan; 60 g). Each child attached the accelerometer (AC) to their waist and wore it from the time they got up in the morning until they went to bed. They were instructed to wear the AC from Monday to Sunday, providing seven consecutive days of data. The accelerometer has been previously validated in the study of children and adolescents [25,26,27]. According to the scale, distinctions were made regarding intensity (levels 1–3, 4–6, and 7–9), where light PA (LPA) was defined as AC intensity levels of 1–3 and 1.5–2.9 metabolic equivalent of task; moderate PA (MPA): AC intensity levels of 4–6 and 3.0–5.9 METs; vigorous PA (VPA): AC intensity levels of 7–9 and ≥6.0 METs [28,29]. We recorded crude step counts to estimate activity levels, and the time spent in MVPA (≥3.0 METs) was calculated as the sum of the MPA and VPA minutes, and TPA = LPA + MPA + VPA for each day. The study required at least four days of recording (including weekends), with a minimum of 10 h of wear time per day for inclusion in the subsequent analysis [30]. We excluded days when no signal was detected by the AC for more than one hour, with this period regarded as non-wearing time.

### 2.5. Data Analysis

The results are presented as mean and standard deviation (SD). We conducted a two-way analysis of variance (ANOVA) to determine the variability in each muscle thickness and daily PA outcome (steps, MVPA, and TPA) among the sexes, and weight status. A 2 (sex: boys and girls) × 2 (weight status: normal and overweight/obese) ANOVA was used to assess variability during weekdays and weekends. ANOVA with Bonferroni post hoc test was used to determine their significance. 

To determine the bivariate relationships between all indicators of muscle thickness and daily PA, we conducted linear regression models (reported as Pearson’s correlation coefficients) by inserting all indicators of muscle thickness as independent variables and daily PA as dependent variables. Two models were used in this study. Model 1 was an unadjusted analysis of the muscle thickness, fat, and PA. Model 2 was adjusted for height, weight, and duration of daylight. There was no interaction noted between monthly age groups and the PA outcomes, which indicates that the pattern of the association with muscle thickness and daily PA was similar in 3- and 4-year-old preschool children. Therefore, the analysis grouped children aged three and four years together and included the monthly age as a covariate in the model.

Statistical analysis was conducted using SPSS software (version 22.0; SPSS Inc., IBM, NY, USA: IBM Corp.), and the level of significance was set at *p* < 0.05. 

The study report followed the Strengthening the Reporting of Observational Studies in Epidemiology (STROBE) Statement (Appendix A).

## 3. Results

### 3.1. Participant Characteristics

A total of 86 preschool children (47 boys and 39 girls) met the inclusion criteria and were included in the analyses (Table 1). For preschool children, the median age was 49 ± 6 months for boys and girls, as shown in Table 2. Boys and girls had similar physical characteristics. Of the study population, 13.8% of the boys and 8.6% of the girls were classified as overweight or obese according to the WGOC thresholds of BMI. Boys and girls in the weight status groups had significantly higher weights and BMI than those in the NW group (Table A1, *p* < 0.05). The sample selection process is detailed in Figure 1.

### 3.2. Muscle Thickness and Fat Thickness Values

Descriptive characteristics and ultrasound MT measurements of preschool children in relation to sex and weight status are reported in Figure 2. Girls had significantly lower muscle thicknesses in the UA, FA, and AL than boys. For boys, the OW/OB group had significantly greater thickness in the UA, FA, AT, PT, AL, and PL than those in the NW group (*p* < 0.05). For girls, the OW/OB group had significantly greater FA, AT, PT, AL, and PL thickness.

All indicators of fat thickness values were significantly higher in girls than in boys as well as in the OW/OB group compared to their NW peers (*p* < 0.05).

### 3.3. Daily Physical Activity Outcomes

The daily physical activity outcomes for the boys and girls are shown in Table 2. There was no significant difference in steps, MVPA, and TPA between the boys and girls, except for in the TPA on weekends (girls were significantly less active than boys (107.8 min ± 22.4 vs. 124.0 min ± 24.4, *p* < 0.05)).

According to weight status, no significant differences were found in steps, MVPA, and TPA between the NW and OW/OB groups on both the weekdays and weekends in either the boys or girls (Table A2).

### 3.4. Association between Muscle Thickness and Fat Thickness Values in Line with Physical Activity

The results of the linear regression models are shown in Table 3. For boys, the AB muscle was positively associated with daily steps and MVPA on weekdays. The PT muscle was positively associated with steps on both weekdays and weekends and MVPA on weekends (*p* < 0.05). Regarding fat thickness, only FA was negatively associated with MVPA on weekdays (*p* < 0.05).

In girls, the PT and PL muscles were positively associated with daily steps, MVPA, and TPA on weekends. With regards to fat thickness, PT was negatively associated with daily steps and MVPA on weekdays, whereas PL was negatively associated with MVPA on weekdays (*p* < 0.05).

The adjusted associations between muscle thickness and fat thickness with PA for each characteristic measurement are shown in Table 4. In the adjusted analyses, the most consistent results were established for the PT muscle, which was significantly positively associated with daily steps and MVPA on weekdays and TPA on both weekdays and weekends. Moreover, the fat thickness in the UA was negatively associated with daily steps, MVPA, and TPA on weekdays in boys (*p* < 0.05).

For girls, in the adjusted analyses, the most consistent results were established for the PT and PL muscles, which were significantly positively associated with daily steps and MVPA on weekdays, and MVPA on weekends, respectively. Moreover, AB was positively associated with all PA outcomes on weekdays and weekends. For fat thickness, PT and PL were inversely associated with daily steps and MVPA on weekdays, independent of covariates, respectively (*p* < 0.05).

## 4. Discussion

This study examined the association between muscle thickness and PA using the weight status of Chinese preschool children after adjusting for several potential confounding factors. These findings expand on the current literature since this is, to the best of our knowledge, the first study to report an association between measures of muscle thickness and PA in preschoolers in China. Our data showed that the muscle thicknesses of preschool children had significant positive associations with PA, both in daily steps and MVPA. The relationships were stronger for the lower legs than for the arms, and for girls than for boys.

In this study, after adjusting for monthly age, height, and weight, we observed that the relationships for the muscle thickness of the lower limb were positively associated with daily steps and MVPA on weekdays for boys. Similar relationships were also observed in the AB muscle, and the lower limb was positively associated with daily steps and MVPA on both weekdays and weekends for girls. These results suggest that decreasing muscle thickness and increasing fat thickness in early childhood (OW/OB) may affect fitness performance and PA.

Our results indicated that the positive associations between muscle thickness and the objective measurements of PA, especially a moderate–vigorous intensity, correlated more strongly with lower limb muscle mass than with upper limb muscle mass. In a recent study, positive associations were noted between vigorous physical activity (PA) and the strength of the lower-body muscles in assessments conducted on both adolescents and adults [7]. Few studies have investigated the association between muscle thickness and PA according to weight status in preschool children. There is some evidence that showed a significantly positive relationship between walking speed and knee extension torque with both daily steps and intensity of activity >3 METs [31]. These observations may indicate a causal relationship, since walking, the primary component of our indices of habitual activity, is more likely to maintain muscle function in the legs than in the arm [32]. Additionally, even after accounting for various potential confounding factors, the associations we observed with objectively measured PA levels remained statistically significant. Recent meta-analyses focusing on PA and metabolic outcomes have shown that a mere 10 min increment in MVPA is linked to reduced waist circumference and lower fasting insulin levels [11,33]. To our knowledge, no previous study has reported an association between muscle thickness and PA in preschool children. Thus, in the present study, the lack of daily steps and MVPA may be major contributing factors to the increasing prevalence of overweight or obese children.

Our findings showed a significant effect of sex on all indicators of fat thickness, yet no difference in muscle thickness. Cross-sectional studies have shown consistent gender-related variations in muscle thickness among children aged 3–18 years, both in boys and girls [2], although the differences were only observed in the upper arm. In Western countries, the percentage body fat rate rapidly increases to 25% one year after birth, then, decreases around the age of 5 years. Girls begin to increase in body fat percentage around the age of 6 years, and it continues to increase until adulthood. Boys, however, only begin to increase their body fat percentage from the age of 7 years and peak at 11 years old, after which a decrease is observed due to the rapid increase in fat-free mass [3]. Therefore, our findings affirm the outcomes of previous research, underscoring that boys generally exhibit significantly lower body fat levels than girls. Conversely, this study reveals that boys tend to have notably greater muscle thickness in certain ranges compared to girls [34,35]. In our study, we did not observe significant gender disparities in BMI between boys and girls, which is consistent with findings from previous research [35]. The sex-specific pattern of fat accumulation during childhood was characterized by a steady increase in subcutaneous fat accumulation in girls. From these considerations, it was shown that sex differences do exist in the fat thickness of the upper and lower limbs from infancy to early childhood. 

Our results also revealed no significant differences in PA between weekdays and weekends among NW and OW/OB children in these age groups, although the intensity level of PA was higher in the group of preschool children who were not overweight/obese. Comparable information was disclosed in a Portuguese study involving preschool children in the same age group [36]. On the contrary, numerous studies conducted in older children (aged > 6 years) have demonstrated a strong and statistically significant inverse connection between weight status and PA in overweight and obese children, who engaged in notably less PA compared to their normal weight counterparts [22,37]. Extending these findings in children, a recent study revealed a close relationship between the body fat percentage and the level of PA, whereby children with a higher percentage of body fats were less physically active, both in terms of steps per day and moderate to vigorous physical activity [20]. Moreover, a recent study has furnished evidence that a mere 10 min increase in daily moderate or vigorous physical activity (PA) can result in an average elevation of 1–2% in bone stiffness on a substantial sample of children aged 2–10 years [10]. The fact that we did not find a correlation between weight status and daily PA in preschool children leads us to conclude that, in younger children, the natural drive to be active is not influenced by a higher BMI, as it seems to be in old children. Therefore, it is essential to initiate preventive interventions in this age group. The urgent need for early intervention is underlined by the fact that PA seems to decrease from childhood to adolescence. As there are no validated obesity prevention strategies for childhood and adolescence, early identification of potential risk factors is crucial. In adults, the level of PA may predict the risk of developing chronic diseases in the future. However, our results suggest that risk prediction for obesity and associated diseases may not be assumed by PA in preschool children, but PA may be of increasing importance during the course of childhood development [38]. Thus, promotion of PA that starts as early as possible may prevent chronic diseases later in life.

According to sex and the day of the week (weekdays and weekends), this study did not identify differences in PA between sex and the day of the week. This is consistent with some [39,40], but not all previous studies [41,42]. Previous research has suggested that the differences in sex and the day of the week might be explained by the fact that PA in preschool children is not greatly influenced by the environment, but rather by personal factors [39]. Rowlands et al. [43] found that PA decreased in children aged 9 to 11 years during weekends. These findings suggest that the tendency of children to be less active during weekends begins at a very young age, which might require special attention in future activity interventions. Moreover, we also believe that issues, such as the involvement of family and the perception of an unfavorable family environment, together with social roles for girls, may play a role in why children are less active [44] over the weekend.

This study has some limitations. Since our sample only included preschool children from a large metropolitan area, it makes it difficult to generalize these findings. Accelerometers also possess certain limitations when it comes to evaluating overall activity levels. It is crucial to acknowledge that, particularly for preschool children, a shorter accelerometer interval (e.g., 5 s) is strongly advised due to the sporadic nature of their physical activity [45]. As a consequence, there is a potential for underestimating sedentary time and vigorous PA, while moderate PA may be overestimated due to the shorter accelerometer interval recommended for preschool children. This aspect should be considered when interpreting data related to physical activity levels for children from an early age. Although we controlled for several potential confounders, such as age, height, and weight, we cannot be certain that other unmeasured confounders, such as dietary intake or genetic variation, did not have an influence on our findings. Therefore, future studies should address these issues.

## 5. Conclusions

In conclusion, our study demonstrated that higher muscle and lower fat thickness were associated with higher levels of PA, especially MVPA. The findings also suggest that preschool children seem to be engaging adequately in PA. Developing health habits at this early stage increases the probability of these habits continuing into later life. It may be as important to maintain and/or increase the amount of PA in preschool children as it is to foster awareness of the importance of PA and transform it into a regular daily habit.

## Figures and Tables

**Figure 1 children-10-01538-f001:**
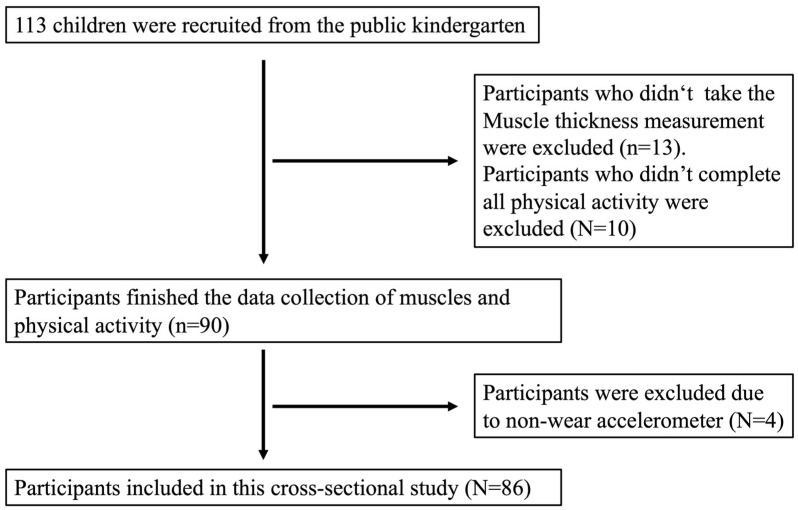
STROBE study flow of participants.

**Figure 2 children-10-01538-f002:**
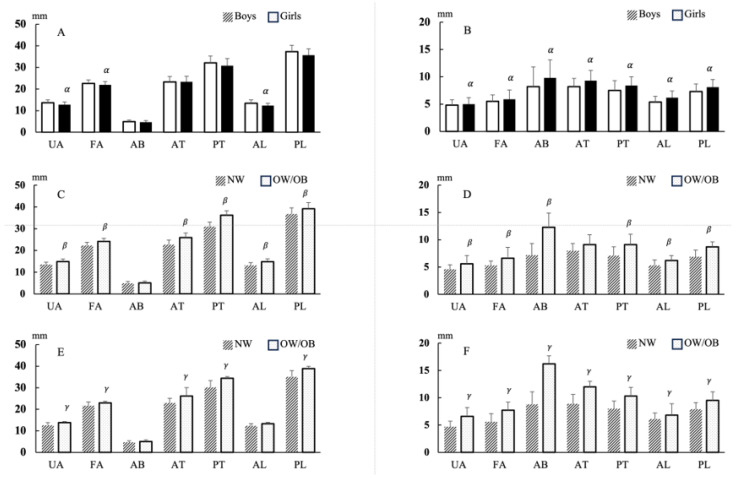
Muscle thickness and fat thickness outcomes of preschool children. (**A**) Muscle thickness of boys and girls; (**B**) Fat thickness of girls; (**C**) Muscle thickness between NW and OW/OB boys; (**D**) Fat thickness between NW and OW/OB boys; (**E**) Muscle thickness between NW and OW/OB girls; (**F**) Fat thickness between NW and OW/OB girls. NW: normal weight; OW/OB: overweight/obese; UA: upper arm; FA: forearm; AB: abdomen; AT: anterior thigh; PT: posterior thigh; AL: anterior lower leg; PL: posterior lower leg. *α* vs. boys *p* < 0.05; *β* vs. NW boys *p* < 0.05; *γ* vs. NW girls *p* < 0.05.

**Table 1 children-10-01538-t001:** Physical characteristics of preschool children.

ALL ^a^			
	Boys	Girls	Sex difference
	N = 47	N = 39	
Age (month)	49 ± 6	49 ± 6	
Height (cm)	106.6 ± 5.0	103.9 ± 5.3	0.257
Weight (kg)	17.9 ± 2.5	16.7 ± 2.3	0.370
BMI	15.7 ± 1.4	15.4 ± 1.3	0.837
Boys ^a^			
	NW	OW/OB	Body type difference
	N = 38	N = 9	
Age (month)	50 ± 6	51 ± 6	
Height (cm)	106.2 ± 5.0	108.4 ± 4.6	0.249
Weight (kg)	17.3 ± 1.8	20.7 ± 3.1	*p* < 0.05
BMI	15.3 ± 1.0	17.5 ± 1.3	*p* < 0.05
Girls ^a^			
	NW	OW/OB	Body type difference
	N = 34	N = 5	
Age (month)	49 ± 7	50 ± 3	
Height (cm)	103.4 ± 5.4	107.7 ± 2.9	0.072
Weight (kg)	16.1 ± 1.8	20.7 ± 0.7	*p* < 0.05
BMI	15.0 ± 0.9	17.9 ± 0.8	*p* < 0.05

Note: Mean ± SD; NW: normal weight; OW/OB: overweight/obese; BMI: body mass index. ^a^ Tested with *t*-test.

**Table 2 children-10-01538-t002:** Daily physical activity outcomes of boys and girls.

	Boys	Girls	Sex Difference	Daily Difference
Weekday				
Daily steps (steps/day)	12,185 ± 2549	10,808 ± 2411	0.058	0.359
MVPA (min/day)	39.8 ± 11.2	36.5 ± 12.0	0.482	0.418
TPA (min/day)	124.0 ± 24.4	107.8 ± 22.4	*p* < 0.05	0.535
Weekend				
Daily steps (steps/day)	11,632 ± 4006	10,848 ± 5038	0.465	
MVPA (min/day)	39.5 ± 18.1	36.7 ± 24.0	0.651	
TPA (min/day)	115.9 ± 39.0	108.2 ± 46.3	0.418	

Note: Mean ± SD; MVPA: moderate–vigorous physical activity; TPA: total physical activity.

**Table 3 children-10-01538-t003:** Correlation among muscle thickness, fat thickness, and physical activity for boys and girls ^a^.

		Boys						Girls					
		Weekday			Weekend			Weekday			Weekend		
		Daily Steps	MVPA	TPA	Daily Steps	MVPA	TPA	Daily Steps	MVPA	TPA	Daily Steps	MVPA	TPA
Muscle Thickness	UA	0.13	0.02	0.15	0.07	0.04	0.13	−0.18	−0.04	−0.20	0.15	0.20	0.13
	FA	0.12	0.01	0.14	−0.04	0.01	0.208	−0.12	−0.09	−0.08	0.19	0.21	0.19
	AB	0.32 *	0.37 *	0.32	0.15	0.21	0.14	0.28	0.19	0.31	0.27	0.31	0.25
	AT	0.24	0.16	0.27	0.11	0.06	0.16	0.10	0.02	0.14	0.22	0.15	0.22
	PT	0.32 *	0.20	0.35	0.17	0.12	0.23	0.10	0.08	0.12	0.38 *	0.35 *	0.39 *
	AL	−0.08	−0.13	−0.03	−0.14	−0.05	−0.13	−0.01	0.09	−0.02	0.18	0.23	0.16
	PL	0.09	0.02	0.14	0.02	0.02	0.06	0.18	0.22	0.16	0.43 *	0.44 *	0.43 *
Fat Thickness	UA	−0.13	−0.20	−0.12	−0.06	−0.03	−0.06	−0.01	−0.17	0.04	−0.01	−0.11	0.03
	FA	−0.18	−0.28 *	−0.14	−0.19	−0.12	−0.18	−0.09	−0.11	−0.09	0.22	0.18	0.24
	AB	0.04	−0.09	0.09	0.01	−0.06	0.05	−0.07	−0.17	−0.02	0.16	0.06	0.20
	AT	−0.14	−0.20	−0.11	−0.09	−0.06	−0.07	−0.27	−0.25	−0.23	0.06	0.02	0.10
	PT	−0.05	−0.22	0.01	−0.12	−0.07	−0.11	−0.35 *	−0.41 *	0.30	−0.03	−0.10	0.03
	AL	0.09	0.01	0.14	0.07	0.08	0.08	−0.28	−0.23	−0.27	−0.02	−0.03	−0.01
	PL	−0.15	−0.26	−0.07	−0.14	−0.09	−0.13	−0.31	−0.36 *	−0.28	−0.02	−0.10	0.03

Note: UA: upper arm; FA: forearm; AB: abdomen; AT: anterior thigh; PT: posterior thigh; AL: anterior lower leg; PL: posterior lower leg; MVPA: moderate–vigorous physical activity; TPA: total physical activity; * *p* < 0.05. ^a^ Model 1 was an unadjusted analysis of the muscle thickness, fat thickness, and PA.

**Table 4 children-10-01538-t004:** Correlations among muscle thickness, fat thickness, and physical activity for boys and girls, controlling for years, height, and weight ^b^.

		Boys						Girls					
		Weekday			Weekend			Weekday			Weekend		
		Daily Steps	MVPA	TPA	Daily Steps	MVPA	TPA	Daily Steps	MVPA	TPA	Daily Steps	MVPA	TPA
Muscle Thickness	UA	0.07	0.05	0.04	0.15	0.16	0.17	−0.14	−0.01	−0.20	0.14	0.19	0.11
	FA	0.09	0.03	0.05	0.04	0.11	0.05	−0.05	−0.02	−0.06	0.16	0.17	0.14
	AB	0.16	0.27	0.18	0.07	0.20	0.01	0.50 *	0.41 *	0.48 *	0.41 *	0.46 *	0.39 *
	AT	0.10	0.10	0.09	0.18	0.16	0.19	0.21	0.23	0.20	0.28	0.27	0.27
	PT	0.39 *	0.34 *	0.35 *	0.28	0.26	0.31 *	0.41 *	0.36 *	0.40	0.34	0.37 *	0.29
	AL	−0.17	−0.09	−0.18	−0.12	0.04	−0.17	0.06	0.23	−0.01	0.08	0.19	0.02
	PL	−0.18	−0.08	−0.01	0.06	0.01	−0.06	0.50 *	0.57 *	0.44 *	0.39 *	0.49 *	0.33
Fat Thickness	UA	−0.33 *	−0.38 *	−0.35 *	−0.12	−0.08	−0.14	−0.03	−0.05	−0.01	−0.20	−0.18	−0.18
	FA	−0.33 *	−0.35 *	−0.34 *	−0.22	−0.09	−0.26	−0.09	−0.13	−0.09	0.28	0.28	0.29
	AB	−0.10	−0.13	−0.12	0.01	−0.05	0.02	0.01	−0.11	0.04	0.08	−0.01	0.12
	AT	−0.22	−0.23	−0.22	−0.12	−0.08	−0.12	−0.26	−0.31	−0.22	−0.01	−0.05	0.02
	PT	−0.15	−0.03	−0.13	0.13	−0.04	−0.16	−0.48 *	−0.49 *	−0.45 *	−0.09	−0.14	−0.05
	AL	0.12	0.06	0.16	0.04	0.05	0.05	−0.26	−0.30	−0.23	0.06	−0.01	0.08
	PL	−0.21	−0.26	−0.17	−0.11	−0.05	−0.13	−0.39 *	−0.43 *	−0.35	−0.07	−0.15	−0.03

Note: UA: upper arm; FA: forearm; AB: abdomen; AT: anterior thigh; PT: posterior thigh; AL: anterior lower leg; PL: posterior lower leg; MVPA: moderate–vigorous physical activity; TPA: total physical activity; * *p* < 0.05. ^b^ Model 2 was adjusted for height, weight, and duration of daylight.

## Data Availability

The data presented in this study are available on request from the corresponding author. The data are not publicly available due to privacy restrictions.

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
