# Peer review of "Associations of Morphological Changes in Skeletal Muscles of Preschool Children in China Following Physical Activity"

_children, 2023, doi:10.3390/children10091538_

Round 1

Reviewer 1 Report

I recommend changing the objective to “assess whether associations exist between x and y”. "Evaluating association" presupposes its existence.

I recommend using the STROBE checklist to support the report. I point out weaknesses in the reports on sampling and losses, which are essential to the internal validity of the research.

Apart from the importance of the topic and the objectives of the study, I consider that 8 tables are excessive. I recommend reviewing the number of tables, bringing to the body of the manuscript only those that contain the most important information, directly related to the conclusions. The remaining tables can be presented as supplementary material.

I disagree with pointing out the cross-sectional design as a limitation. Within what the authors propose as an objective, it is the best design for the study. What should be reinforced is care in interpreting the result. Association does not represent causation, and there is also the possibility of reverse causality.

Author Response

Dear Reviewer,

Despite your busy schedule, thank you very much for your valuable feedback. We have studied comments carefully and have made correction which we hope meet with approval. We tried our best to improve the manuscript and made some changes in the manuscript. These changes will not influence the content and framework of the paper. Note:Red marks the revision.

Despite your busy schedule, thank you very much for your valuable feedback. We have studied comments carefully and have made correction which we hope meet with approval. We tried our best to improve the manuscript and made some changes in the manuscript. These changes will not influence the content and framework of the paper. Note:Red marks the revision.

  1. I recommend changing the objective to “assess whether associations exist between x and y”. "Evaluating association" presupposes its existence.

Response: I concur with your recommendation. Modifying the objective to 'assess whether associations exist between site-specific muscle changes and PA' is justifiable. The term 'evaluating association' is more precise and avoids assuming the presence of an association, aligning better with scientific research standards. We appreciate your suggestion and will implement this change (L25-27, L112-113).

  1. I recommend using the STROBE checklist to support the report. I point out weaknesses in the reports on sampling and losses, which are essential to the internal validity of the research.

Response: Thank you for your suggestion; I appreciate it. I concur that utilizing the STROBE checklist to bolster our report is an excellent idea. This approach will contribute significantly to maintaining the quality and transparency of our research. Furthermore, I value your emphasis on the significance of addressing deficiencies related to sampling and losses, as they play a pivotal role in ensuring the internal validity of our study. We are committed to integrating the STROBE checklist and conducting a comprehensive evaluation to enhance these aspects of our report, ultimately improving its overall quality (Figure 1).

  1. Apart from the importance of the topic and the objectives of the study, I consider that 8 tables are excessive. I recommend reviewing the number of tables, bringing to the body of the manuscript only those that contain the most important information, directly related to the conclusions. The remaining tables can be presented as supplementary material.

Response: Response: Thank you for pointing it out; I appreciate your valuable input. I concur that including 8 tables might be excessive, and it is essential to maintain the manuscript's focus on the most relevant information pertaining to our conclusions. I will conduct a comprehensive review of the tables and choose only those that contain essential data directly linked to our study's outcomes for inclusion in the main body of the manuscript (Figure 1-2, Table 1-4). The remaining tables will be presented as supplementary material, ensuring the main text's clarity and relevance (Addition file 1-3).

  1. I disagree with pointing out the cross-sectional design as a limitation. Within what the authors propose as an objective, it is the best design for the study. What should be reinforced is care in interpreting the result. Association does not represent causation, and there is also the possibility of reverse causality.

Response: Thank you for sharing your viewpoint. I understand your perspective regarding the cross-sectional design, and I agree that it can indeed be the most suitable choice within the scope of our stated objectives. I appreciate your emphasis on the importance of careful result interpretation. You are absolutely correct in noting that association does not imply causation, and the possibility of reverse causality should always be considered. We will ensure that our paper emphasizes these points to avoid any misinterpretation of the findings and to provide a clear understanding of the limitations and strengths of our study design (L488-496).

Reviewer 2 Report

First of all, thank the journal for the invitation to review the paper. I have read it very carefully, since the evaluation of the physical activity of children of these ages is my specialty. The paper is very interesting and adds evidence in a field of physical activity research that needs much more research.

Therefore, I encourage the authors to continue on this path. Another interesting element in the research is the objective evaluation of the levels of physical activity with an accelerometer, there are works in this sense, but they use indirect and often subjective evaluation systems that introduce biases in the investigation and that the inference of the conclusions. I think it is necessary to highlight in this investigation the evaluation tools used, many of which are standard goals. Finally, in this general evaluation of the paper, the application and transfer must be highlighted, in the sense that it is necessary to increase the levels of physical activity in the early stages of childhood.

On page 3 the authors say "index (BMI) was calculated using the following standard equation: BMI= weight in kg/height in meters" It is not necessary to report on this, since this formula is fully known by specialists who read this article. In my opinion, it is not necessary to indicate the formula.

In my opinion, the evaluation of Muscle thickness (page 3) is well detailed, and as I said before, it seems to me an adequate and pertinent protocol for this investigation. On the other hand, the accelerometer used is not a standard in the evaluation of PA, but the protocol is well detailed, with which the results can be optimally analyzed.

In relation to the results, they coincide, for example, with the work of the Portuguese professor Dr. Jorge Mota, who is a worldwide reference and supports the results of this study (reference 35).

Once the paper has been analyzed, I recommend its publication as I believe that this type of study can help to better understand the characteristics of PA in children and design strategies to prevent the obesity pandemic.

Author Response

Dear Reviewer,

I will send you the revised paper all at once. I would like to express my gratitude to you for taking the time to review my manuscript. I appreciate the careful consideration and useful comments provided. Despite your busy schedule, thank you very much for your valuable feedback.

Reviewer 3 Report

Congratulations to the authors for the study.

The approach of the study is based on a correct and accurate consideration, although a number of clarifications are needed.

Abstract error, a parenthesis is missing after PA).

Are so many tables of results really necessary? They add essential information to the article.

The discussion uses a large number of citations that have not been taken into account in the introduction. This should be reconsidered, as the introduction is where the main ideas should be presented, taking into account the most important references that will later be used in the discussion.

The scarce use of recent quotations in the text is striking. How do they justify the current interest in this type of study in this case? Likewise, the quotations used in the methodology section are not recent either, as the measuring instruments are constantly evolving.

The first reference has lost its format in the text.

Author Response

Dear Reviewer,

Despite your busy schedule, thank you very much for your valuable feedback. We have studied comments carefully and have made correction which we hope meet with approval. We tried our best to improve the manuscript and made some changes in the manuscript. These changes will not influence the content and framework of the paper. Note:Red marks the revision.

  1. Abstract error, a parenthesis is missing after PA).

Response: We have made the necessary corrections to the abstract, as you pointed out.

  1. Are so many tables of results really necessary? They add essential information to the article.

Response: Thank you for pointing it out; I appreciate your valuable input. I concur that including 8 tables might be excessive, and it is essential to maintain the manuscript's focus on the most relevant information pertaining to our conclusions. I will conduct a comprehensive review of the tables and choose only those that contain essential data directly linked to our study's outcomes for inclusion in the main body of the manuscript (Figure 1-2, Table 1-4). The remaining tables will be presented as supplementary material, ensuring the main text's clarity and relevance (Addition file 1-3).

  1. The discussion uses a large number of citations that have not been taken into account in the introduction. This should be reconsidered, as the introduction is where the main ideas should be presented, taking into account the most important references that will later be used in the discussion.

Response: Thank you for pointing it out; You are accurate in highlighting the inconsistency between the citations in the discussion section and those introduced in the introduction. We acknowledge the importance of the introduction in establishing the foundation for the main ideas, encompassing the most pivotal references that will be further elaborated upon in the discussion. To address this concern, we were to meticulously review and revise the introduction, integrating the essential references that hold significance to our subsequent discussion. This modification aims to enhance the cohesion and coherence of our paper by ensuring a better alignment between the introduction and the ensuing discussion sections." (L45-52, L58-59, L74-76, L281-283, L289-292, L298-310, L318-328)

  1. The scarce use of recent quotations in the text is striking. How do they justify the current interest in this type of study in this case? Likewise, the quotations used in the methodology section are not recent either, as the measuring instruments are constantly evolving.

Response: Thank you for bringing up the issue of limited recent citations in our study. We understand your concern and would like to clarify our approach. While it's true that our citations may not be the most recent, our intent was to establish a robust foundation for our study by including seminal works and key references that have laid the groundwork for the current state of research in this area. However, as you pointed out, especially when discussing methodology section, we have made the correction to include more recent citations.

  1. The first reference has lost its format in the text.

Response: As you pointed out, we have made the necessary corrections to the references.

Round 2

Reviewer 1 Report

Dear authors, 

The changes made qualified the material. Congratulations on the progress.

Reviewer 3 Report

Congratulations to the authors for their work in improving the article.